# New Insights on the Feature and Function of Tail Tubular Protein B and Tail Fiber Protein of the Lytic Bacteriophage φYeO3-12 Specific for *Yersinia enterocolitica* Serotype O:3

**DOI:** 10.3390/molecules25194392

**Published:** 2020-09-24

**Authors:** Anna Pyra, Karolina Filik, Bożena Szermer-Olearnik, Anna Czarny, Ewa Brzozowska

**Affiliations:** 1Faculty of Chemistry, University of Wroclaw, 14 F. Joliot-Curie St, 50383 Wroclaw, Poland; 2Hirszfeld Institute of Immunology and Experimental Therapy, Polish Academy of Sciences, 12 R. Weigl St, 53114 Wroclaw, Poland; karolina.filik@hirszfeld.pl (K.F.); bozena.szermer-olearnik@hirszfeld.pl (B.S.-O.); anna.czarny@hirszfeld.pl (A.C.)

**Keywords:** tail tubular proteins, tail fiber proteins, biofilm

## Abstract

For the first time, we are introducing TTPBgp12 and TFPgp17 as new members of the tail tubular proteins B (TTPB) and tail fiber proteins (TFP) family, respectively. These proteins originate from *Yersinia enterocolitica* phage φYeO3-12. It was originally thought that these were structural proteins. However, our results show that they also inhibit bacterial growth and biofilm formation. According to the bioinformatic analysis, TTPBgp12 is functionally and structurally similar to the TTP of *Enterobacteria* phage T7 and adopts a β-structure. TFPgp17 contains an intramolecular chaperone domain at its C-terminal end. The N-terminus of TFPgp17 is similar to other representatives of the TFP family. Interestingly, the predicted 3D structure of TFPgp17 is similar to other bacterial S-layer proteins. Based on the thermal unfolding experiment, TTPBgp12 seems to be a two-domain protein that aggregates in the presence of sugars such as maltose and N-acetylglucosamine (GlcNAc). These sugars cause two unfolding events to transition into one global event. TFPgp17 is a one-domain protein. Maltose and GlcNAc decrease the aggregation temperature of TFPgp17, while the presence of N-acetylgalactosamine (GalNAc) increases the temperature of its aggregation. The thermal unfolding analysis of the concentration gradient of TTPBgp12 and TFPgp17 indicates that with decreasing concentrations, both proteins increase in stability. However, a decrease in the protein concentration also causes an increase in its aggregation, for both TTPBgp12 and TFPgp17.

## 1. Introduction

Phage therapy is a promising alternative to antibiotics that are becoming increasingly less effective. Many phages that are effective against pathogenic strains have already been isolated. More than 95% of all phages belong to the *Caudovirales* of order, which includes the *Myoviridae*, *Podoviridae* and *Siphoviridae* families [1]. The bacteriophage φO3-12 belongs to the *Podoviridae* family and it is classified as part of the T7 group [2]. It was reported that most of the predicted gene products of phage φYeO3-12 are over 70% identical to those of the T3 genome. Therefore, the φYeO3-12 phage genome is extensively similar to the *Enterobacteria* T3 and T7 genomes [2]. 

*Yersinia enterocolitica* is a Gram-negative bacterium from the *Enterobacteriaceae* family. It is a widespread bacterium that can be found in soil and water environments. The reservoir of *Yersinia enterocolitica* serotypes is mainly found in pigs. People can become infected by eating uncooked pork contaminated with bacteria or through contact with the feces of livestock [3,4,5,6]. It is one of three known species of *Yersinia*, that is both pathogenic to humans and animals. *Y. enterocolitica* usually causes infections specific to the gastrointestinal track. Some common symptoms of infection are watery diarrhea, fever, vomiting, etc. [7,8].

The bacterial cell wall of *Yersinia*, as with other members of the whole Gram-negative bacteria group, contains an outer membrane in which the major component is lipopolysaccharide (LPS). LPS consists of three main parts: the lipid A, the core oligosaccharide that can often be divided into an inner and outer core, and an O-specific polysaccharide [9]. The presence of the complete LPS is required for the full virulence of *Yersinia enterocolitica* [10]. The outer core (OC) of the LPS is a particularly important virulence factor in *Yersinia enterocolitica* serotype O:3 (YeO3) [11].

The OC saccharide of YeO3 LPS contains 2-acetamido-2,6-dideoxy-d-*xylo*-hex-4-ulopyranose (Sugp), two glucose (Glc), one galactose (Gal) and two *N*-acetylgalactosamine (GalNAc) residues [9]. It was reported that the external part of the LPS also functions as a bacteriophage receptor [9,12]. Therefore, phages, that display specificity for a particular bacterial strain, could be used as potential diagnostic or therapeutic tools.

Phages attach to their host bacteria with the end of their tails. During infection, DNA is ejected from the capsid through a tail complex to the bacterial host. This stage is well characterized for the T7 phage [13]. The tail machine is composed of the following proteins: the portal (gp8), the adaptor (gp11), the nozzle (gp12, T7_TTPgp12) and the fiber (gp17, T7_TFPgp17). The portal protein, which exists in two conformations, acts as a valve on the portal pore. Thus, it is responsible for the regulation of the passage of DNA passage into the capsid. The adaptor protein interacts with the bottom of the portal protein, allowing DNA to slip along the tail up to the nozzle protein. The nozzle protein, T7_TTPgp12, contains four domains: the platform (interacting with the adaptor protein), the fiber dock interaction domain, the central β-propeller domain and the most distal nozzle tip domain. The nozzle, T7_TTPgp12, is the main protein responsible for closing and securing DNA inside the tail of the mature virus. The tail fiber protein, T7_TFPgp17, interacts with the bacterial LPS [14]. It has been reported that the C-terminal receptor-binding domain of T7_TTPgp17 is responsible for host-range determination, most likely by binding to a specific LPS region that varies between bacterial strains [15]. Additionally, it is well known that phages have evolved to be able to infect different hosts by adapting the receptor domain of TFP [16]. The adaptation is propelled by gene transfer, which accidentally leads to the acquisition new features. One example is the intramolecular chaperone (classified as S74 peptidase family) which contains a conserved C-terminal domain of an endosialidase found in the φK1F phage. The gene encoding the chaperone has become a common feature of other TFPs and it can be removed by a self-cleaving mechanism [17,18]. The first 175 N-terminal amino acid residues of this endosialidase show homology to the N-terminal regions of T7_TFPgp17, TFP of T3 phage (T3_TFPgp17) and TFP of φYeO3-12 phage [17]. The C-terminals of the TFPs differ since the C-terminals form the distal part of TFPs, which bind to the host receptors [19].

In our previous paper, we reported the biochemical features of TTPAgp11 of *Yersinia* phage φYeO3-12, which, in addition to its structural function, also acts as a sugar hydrolyzing enzyme [20]. In this paper, we are presenting our results of our bioinformatic analysis, biochemical characterization and antibacterial tests of two phage proteins. These proteins are the tail tubular protein B (TTPBgp12) and tail fiber protein (TFPgp17) originating from the *Yersinia* phage φYeO3-12. First, the nano differential scanning fluorimetry (nanoDSF) was used to assess the folding and stability of both proteins. Moreover, the stability in presence of different saccharides molecules was measured. 

## 2. Materials and Methods 

### 2.1. Gene Cloning and Protein Overexpression, Purification and Analysis

The complete genome of bacteriophage φYeO3-12 was found in GenBank under accession number AJ251805. The genes that encode the tail tubular protein B (TTPB) and the tail fiber protein (TFP) (the gp12 and gp17), were amplified by polymerase chain reaction (PCR) using the following primers: TTPBgp12_FW, TACTTCCAATCCAATGCCATGCGCTCTTATGAGATGAAC and TTPBgp12_RV, TTATCCACTTCCAATGTTATTAGCGGTTAAGTAGACCAGAGG, TFPgp17_FW, TACTTCCAATCCAATGCCATGGCTACAACTATTAAGACCG and TFPgp17_RV, TTATCCACTTCCAATGTTACTAAGTCTTGTCCTTCTCCAAC. The genomic DNA was isolated from the phage lysate using a viral DNA extraction kit (Biocompare) and served as a template (100 ng) in the PCR reaction. A ligation-independent cloning method was used to clone the PCR products into the pMCSG9 vector using a T4 DNA polymerase [21]. Gene overexpression, protein isolation and purification were performed according to a slightly modified procedure described by Pyra et al. [20]. *E. coli* BL21(DE3)pLysS competent cells were used to overexpress both genes. The competent cells were added to Luria-Bertani (LB) medium with ampicillin, as well as chloramphenicol, in working concentrations of 100 µg/mL and 25 µg/mL, respectively. Both proteins were purified using two rounds of nickel-affinity chromatography. Additionally, gel filtration using a Protein Arc 6–600 16/60 HR Sec column equilibrated with 50 mM Tris/HCl buffer, pH 8.0, containing 300 mM NaCl, 5% glycerol and 1 mM dithiothreitol, was used for purification. After the second nickel column, the flow-through fraction containing TTPBgp12 or TFPgp17 was precipitated using ammonium sulfate (95% saturation) overnight at 4 °C. It was then pelleted by centrifugation, dissolved in the buffer mentioned above and then applied to a gel-filtration column. The TTPBgp12 or TFPgp17 fractions were collected and precipitated by ammonium sulfate (95% saturation) and their purity was analyzed by 12% SDS-PAGE [22].

The secondary structure of both proteins was analyzed by circular dichroism (CD) spectroscopy. For the CD analysis, the proteins were dissolved in water with a final concentration of ~ 7 µM. CD measurements were performed four times using a 1 mm cuvette. The measurements were taken in the wavelength range of 350 to 190 nm using a J-1000 Circular Dichroism spectrophotometer (Jasco Inc., Easton, MD, USA). These data were interpreted using the K2D3 method, which provided an estimation of the secondary structure of the proteins [23]. The proteins concentration was determined by the BCA method [24].

### 2.2. Protein Folding and Aggregation Tests

The thermal unfolding and protein aggregation experiments were carried out using a Prometheus instrument with nanoDSF and backreflection technology (NanoTemper, South San Francisco, CA, USA), using methods that were previously described [20].

NanoDSF technology allows for the automatic determination of thermal unfolding transition midpoints, Tm (°C). This technology also calculates the onset of unfolding using the transition midpoint and the slope of the unfolding signal. This method monitors unfolding-related tryptophan and tyrosine fluorescence at the emission wavelengths of 330 nm and 350 nm, respectively.

The Prometheus device emits near-UV light at a wavelength that is scattered by aggregated proteins, which causes only non-scattered light to reach the detector. A direct measure of protein aggregation is the result of a reduction in backreflected light.

Both protein unfolding and aggregation tests were performed simultaneously. Then, 10 µL of TTPBgp12 (0.25 mg/mL, 2.78 µM) or TFPgp17 (0.25 mg/mL, 3.6 µM) was placed in the capillary alone or mixed with one of the following sugars (0.15 mg/mL): melibiose, galactose, glucose, maltose, β-lactose, α-lactose, GalNAc and GlcNAc.

The samples were heated to 95 °C, and the fluorescence was monitored throughout the experiment. Data were analyzed with the PR.ThermControl and PR.StabilityAnalysis software packages (Publisher, South San Francisco, CA, USA) [25]. As for the results, melting scans of the studied proteins showed the following: the fluorescence ratio (350/330 nm) profile used to calculate Tm (°C), the first derivative profile showing the initial temperatures of unfolding, aggregation and the occurrence of conformational changes in the protein sample, as well as the scattering profile plotted as attenuation units (mAU) against temperature (°C) indicating the onset temperature of protein aggregation.

### 2.3. Measuring the Effect on Biofilm Formation

One hundred microliters of inoculum, containing 2.2 × 10^7^ CFU/mL of *Yersinia enterocolitica* O:3 was added to each well on 96-round well plates. Then, 10 µL of 0.5 mM of TTPBgp12 or TFPgp17 was added to the test samples. Ten microliters of lysis buffer (300 mM NaCl, 20 mM Tris-HCl pH 8.0, 5% glycerol, 5 mM BME), in which proteins were suspended, was added to the bacterial inoculum and considered as a negative control. As a positive control, 10 µL of 10 mg/mL [26] gentamicin and 10 µL of bacteriophage φYe O3-12 (4.5 × 10^7^ PFU/mL) were used. All samples were triplicated. The 96-round well plates were incubated at 28 °C for 24 h. After incubation, the resulting biofilm was suspended in PBS and plated on agar plates. Agar plates were incubated at 28 °C for 24 h, after this time the number of colonies on the plates was counted. The bacteria layer was also placed on aLab-Tek^®^ Chamber SlideTM System (Thermo Fisher Scientific Inc., Rochester, NY, USA), and incubated at 28 °C for 24 h. Then the slides were rinsed with water, dried, and stained with 1% crystal violet. After 30 min, the slides were rinsed repeatedly with water and then observed under a light Olympus microscope (100× magnification).

## 3. Results and Discussion

### 3.1. Bioinformatics Analysis

While studying the proteins of bacteriophage tails, it was discovered that some structural tail tubular A proteins (TTPAs), of the environmental *Klebsiella* (KP32 and KP34) and *Yersinia* (φYeO3-12) phages, can have dual-functionality [20,27,28]. These proteins were only previously considered to be structural phage tail proteins. Our results have shown that they exhibit a second function—a specific saccharide hydrolytic activity on biofilms formed by bacteria. This specificity can be used as an antibacterial agent with a wide spectrum of applications. So far, we have only examined TTPAs. The main question was whether other structural tail phage proteins, like tail tubular proteins B (TTPBs) or tail fiber proteins (TFPs), could also play an antibacterial role. The uncharacterized TTPB (UniProt code: Q9T105) and TFP (UniProt code: Q9T0Z9) encoded by gene 12 and 17, respectively, of *Yersinia* phage φYeO3-12 (GenBank code: AJ251805) were chosen for this study. Firstly, the primary structure of both selected proteins was analyzed using BLAST [29], Clustal Omega [30] and HHPred [31].

BLAST amino acid sequence analysis [29] of TTPBgp12 showed that it is similar to the TTPBs of phages that infect other types of bacteria with the highest similarity to bacteriophages infecting *Yersinia, Escherichia, Salmonella, Citrobacter, Serratia, Klebsiella, Leclercia and Enterobacteria* (97–99%) (Appendix A). Notably, amino acid sequence similarity was also found between TTPBgp12 and TFP from phages of *Enterobacter* (99–100%). The observed amino acid sequence diversity is likely a reflection of the phage’s adaptations in response to environmental changes and/or bacterial cell variability. Although a lot of proteins with high amino acid sequence similarity to TTPBgp12 were found, only one of them appeared to have been well characterized [13]. This protein was a tail tubular protein of *Enterobacteria* phage T7, T7_TTPgp12, which is a nozzle tail protein. This is the main protein responsible for closing and securing the DNA inside the tail, as well as phage absorption to the host bacteria outer membrane [13,32].

The Clustal Omega tool was used [24] to perform the amino acid sequence alignment of both proteins (Appendix A) which allowed us to determine that TTPBgp12 is 69% identical to T7_TTPgp12.

The HHPred tool [31] confirmed a high similarity of both proteins predicting that TTPBgp12 is largely structurally homologous to T7_TTPgp12 with 100% probability. This means that the overall fold of TTPBgp12 should be the same as in T7_TTPgp12.

The Phyre server [33] analysis showed that TTPBgp12 adopts probably a mainly β-structure (56%) with only six very short α-helical fragments (4%) and some disordered regions (18%). Domain analysis led to the finding of twenty short regions of polypeptide chain of TTPBgp12, which corresponds to some known domains of a different protein families. Although the confidence level does not specifically reflect the accuracy of the model, the level of confidence in the TTPBgp12 domain prediction was very low, about 20% or lower. Therefore, the conclusion was that there was no specific domain found in the structure of TTPBgp12.

The I-Tasser [34] server was used for further bioinformatic analysis of the 3D structure of TTPBgp12. The I-Tasser server modeled the structure by using the structures deposited in Protein Data bank (PDB) as a template. As a result, five three-dimensional models were generated. The structure of T7_TTPgp12, determined by cryo-electron microscopy (PDB code: 6r21), presented us with the best template for TTPBgp12 modelling (Z-score > 1). The best obtained model of theTTPBgp12 structure suggests it adopts mainly a β-structure (Figure 1). In this prediction, all secondary structure elements matched very well between TTPBgp12 and T7_TTPgp12. This is not surprising, given the almost 70% similarity of the amino acid sequence of both proteins. Interestingly, the only difference was seen in the one loop region which lacked two residues (R744–R745) in the T7_TTPgp12 structure. However, it modelled well in the corresponding loop of the predicted TTPBgp12 structure.

To build the 3D model for TTPBgp12, the Swiss-Model server [35] was used. Seven protein templates were found. Six of them were considered to be less suitable for modeling than the first one with the PDB code: 6r21. Thus, the only one model for TTPBgp12 was predicted. The obtained results were in accordance with TTPBgp12 structure predicted by the I-Tasser server. This was not surprising since both servers used the same protein template for model building.

In conclusion, the amino acid sequence and structure prediction analysis showed a high level of primary and tertiary structure similarity between TTPBgp12 and T7_TTPgp12. This suggests that the proteins have similar biochemical features and functions.

Based on the information deposited in the UniProt database, it is known that TFPgp17 contains a S74 peptidase domain at its C-terminus end (539–645 amino acid residues). This C-terminus end might be an intramolecular chaperone [17] removed after auto-proteolysis [18].

The BLAST analysis [29] showed that TFPgp17 had the highest amino acid similarity (99–100%) to TFPs of other phages infecting *Enterobacter*, *Yersinia*, *Serratia*, *Citrobacter*, *Shigella* and *Escherichia.* There were also a lot of phage TFP amino acid sequences with a lower similarity found in the range of 40 to 90% (Appendix A). No similarity was found between TFPgp17 and the TTPAs or the TTPBs. As for TTPBgp12, the amino acid sequence diversity among the TFPs is also reflective of the phage adaptations to environmental changes. Unfortunately, the most similar proteins to TFPgp17, in terms of their amino acid sequence, have not been studied. Therefore, we decided to align the amino acid sequence of TFPgp17 with amino acid sequences of only well biochemically or structurally characterized proteins. The following proteins were chosen for multiple protein sequence alignment analysis by BLAST [23] and Clustal Omega tool [24]: TFP of *Enterobacteria* phage T7 (T7_TFP), TFP of *Enterobacteria* phage T3 (T3_TFP) of tail spike protein of *Enterobacteria* phage K1F (K1F_TSP), L-shaped TFP of Escherichia phage T5 (T5_TFP), side TFP of prophage K12 (K12_TFP) and TFP of *Enterobacteria* phage lambda (L_TFP). The BLAST results revealed that most of the similarities in the amino acid sequences of the selected proteins were observed in the N-terminal regions of TFPgp17 and T7_TFP, as well as T3_TFP and K1F_TSP (Appendix A). T5_TFP, K12_TFP and L_TFP only had very short fragments in their amino acid sequences that were similar to the polypeptide chain of TFPgp17, mainly present in the center and C-end regions (Appendix A). To clarify, the Clustal Omega alignments were performed for two sets of amino acid sequences. The first input included the protein sequences with a higher BLAST alignment (TFPgp17 and T7_TFP, T3_TFP and K1F_TSP). The second set contained the protein sequences from the BLAST analysis with a smaller degree of similarity (TFPgp17 and T5_TFP, K12_TFP and L_TFP). All these analyses found that there is not a high similarity between the primary structures of TFPgp17 and the selected proteins, with similarity in the range of only 25% to 36% (Appendix A).

Interestingly, the N-terminal of all the compared proteins was much more similar than the C-terminal region, even up to ≈78. This was previously reported by Pajunen et al. [2]. This is in agreement with other findings that state that in many phages, the gene regions coding the C-terminus of TFPs evolve faster than other phage genes, as a result of intense host range selection [36]. This is due to the function of TFP that is responsible for binding to host cell receptors. Our analysis using the HHPred tool [31] also demonstrates a great diversity in the amino acid sequences of TFPs. The obtained results suggest that the polypeptide chain of TFPgp17 contains some structurally homologous fragments, ~100 to 200 amino acid residues long, to six phage proteins with a probability of 99% (Appendix A). These proteins include: endo-*N*-acetylneuraminidase (*Enterobacteria* phage K1F), receptor recognition protein (*Salmonella* phage vB_SenMS16), L-shaped tail fiber protein (*Enterobacteria* phage T5), neck appendage protein (*Bacillus* phage GA-1), phiAB6 tail spike (unidentified phage) and tail spike protein (*Acinetobacter* phage vB_AbaP_AS12). The homology of the rest of the hits from the HHPred analysis of TFPgp17 was predicted with a probability of ~60% and lower. Most of the listed proteins were structural tail phage proteins. The neck appendage protein of *Bacillus* phage GA-1 is classified as a chaperone, same as endo-*N*-acetylneuraminidase of *Enterobacteria* phage K1F, which additionally acts as hydrolase. That allows us to speculate that TFPgp17 is mainly a structural tail fiber protein. This protein could play the role of a chaperone protein, a receptor binding protein responsible for the bacterial cell recognition or could even exhibit hydrolytic activity.

The Phyre server analysis [33] found that the best protein templates for TFPgp17 domain prediction (with confidence 99%) are in accordance with the results performed by the HHPred tool. These templates include the following: the neck appendage protein (*Bacillus* phage GA-1), the endo-*N*-acetylneuraminidase (*Enterobacteria* phage K1F) and the L-shaped tail fiber protein (*Enterobacteria* phage T5). All of the previously listed proteins represent chaperone proteins. The obtained results suggest that TFPgp17 could contain a chaperone domain on its C-terminus of its polypeptide chain. It is worth noting that the results obtained by both bioinformatic tools indicated that the acquired domains had a rather low percentage of identity, within the range of 13–24% for HHPred and 18–33% for Phyre analysis. The Phyre server analysis also discovered that TFPgp17 could contain an adhesion domain on its N-terminus and a confidence range of 90–96%. The secondary structure and disorder prediction performed by the Phyre server showed that TFPgp17 probably contains more α-helical elements (36%) than β-strands (25%). In addition, the disordered regions are on a similar level as the ordered helices (36%).

The I-Tasser server [34] was used to further analyze the spatial structure of TFPgp17. It generated five models of TFPgp17 based on the comparison to the protein structures deposited in the PDB. The best template proteins for modelling appeared to be the X-ray structures of endo-*N*-acetylneuraminidase of *Enterobacteria* phage K1F, (K1F_Nase, PDB code: 3gw6) [18] and the L-shaped tail fiber protein of *Enterobacteria* phage T5 (T5_TFP, PDB code: 4uw8) [36]. These two templates served as models for the prediction of the C-terminal of TFPgp17. The best obtained model of TFPgp17 structure suggests it mainly adopts a β-structure (Figure 2). In this prediction, only the C-terminal β-structure elements of TTPBgp12 and T5_TFP matched up well.

Sizeable differences were observed between the rest of the modeled TFPgp17 molecule and the templates proteins. This is not surprising, since the amino acid similarity of these regions, in TFPgp17 and T5_TFP, as well as K1F_Nase, is 15% and 13%, respectively. Therefore, we could assess that the overall structure of TFPgp17 predicted by the I-Tasser server was not correct. Moreover, this server has listed ten structural analogs in PDB to the predicted TFPgp17. The first required structure was a solid fit to the TFPgp17 model (Figure 2). That was the bacterial membrane protein RsaA with PDB code: 5n8p, which represents S-layer proteins [38]. This group of proteins includes a diverse class of molecules found in the surface layer (S-layer) of Gram-negative, Gram-positive bacteria and most archaea. The S-layer consists of repeating molecules of S-layer proteins that protect bacterial cells from the external environment, as well as play a role in pathogenicity. Based on this analysis, the following questions arise: Is it possible that the overall fold of TFPgp17 is actually similar to that of RsaA? Or, is it possible that, given the similar predicted structure of TFPgp17 to RsaA, could the phage protein exhibit similar features as have been seen with RsaA? Thus, could it build into the outer membrane of host bacterial cells, which would help the phage to infect bacteria.

Another result from the Swiss-Model server presented with some interesting new information. In this study, 33 templates from PDB for TFPgp17 amino acid sequences were filtered. The templates with the highest quality, predicted from features of the target–template alignment, were selected for model building. The Swiss-Model server generated three partial models of the 3D TFPgp17 structure (Figure 3).

Model no. 1 was built on the template for adhesin A (PDB code: 3d9x) [39] and contains a fragment of the amino acid, containing residues 128–162, which adopts a mixed α/β secondary structure (37% sequence similarity, 5% coverage). The template for model no. 2 was a neck appendage protein (intramolecular chaperone) of *Bacillus* phage GA-1 (PDB code: 3gud) [18]. The predicted model includes the C-terminal S74 peptidase domain built of 539–640 amino acid residues (34% sequence similarity, 14% coverage). Additionally, the last 3D model of TFPgp17 was predicted for its C-terminal amino acid sequence (residues 521–643) using an L-shaped tail fiber protein from *Enterobacteria* phage T5 (T5_TFP, PDB code: 4uw8) [24] as a template (27% sequence similarity, 16% coverage). Models nos. 2 and 3 consist of a mixed α/β secondary structure, with an abundance of a helical structures.

In summary, all the bioinformatic tools used for the TFPgp17 amino acid sequence analysis allowed us to state that TFPgp17 undoubtedly belongs to the phage tail fiber protein family having an intramolecular chaperone domain at its C-terminal end. The N-terminal end of TFPgp17 showed a much higher similarity with another well-characterized representative of this protein family than its C-terminal regions. The great variability of the C-terminal regions is a common feature of TFPs due to being responsible for binding to the host receptors. The partially predicted TFPgp17 3D structure showed that this protein could possess an adhesion domain at the N-terminal region and an autoproteolytic S74 peptidase domain at the C-terminus of its polypeptide chain. The 3D structure of the whole TFPgp17 molecule was also predicted. It demonstrated that this protein could adopt an overall fold very similar to RsaA, which is reported as being a bacterial S-layer protein providing mechanical stability of the cell and protecting bacteria from the outside conditions. For a better characterization of biochemical features of TTPBgp12 and TFPgp17, experimental studies were carried out.

### 3.2. Gene Cloning and Protein Overexpression, Purification and Analysis

The genes for TTPBgp12 and TFPgp17 were cloned into a pMCSG9 vector and both overexpressed proteins were purified in two rounds of nickel-affinity chromatography. This was followed by gel filtration and analyzed with 12% SDS-PAGE (Figure 4 and Figure 5).

The SDS-PAGE showed that the purity of both proteins was over 90% and indicated that the molecular weight of TTPBgp12 was ~90 kDa. This is in accordance with the theoretical value. It is interesting that the theoretical molecular mass of TFPgp17 is 69.44 kDa. However, based on the SDS-PAGE, the molecular weight was determined to be ~58 kDa. This was not surprising, since TFPgp17 is a protein containing a S74 peptidase domain, which is responsible for protein autolysis. Hence, the molecular weight of the MBP-TFPgp17 fusion protein was ~110 kDa. However, after TEV protease cleaved off the fusion protein and the second round of Ni^2+^-affinity chromatography, the molecular weight of TFPgp17 was indicated to be ~58 kDa with SDS-PAGE. During this step of purification, a small protein, ~11 kDa, was also observed in the gel. This corresponded to the molecular mass of the C-terminal S74 peptidase domain (539–645 amino acid residues).

### 3.3. Protein Folding and Aggregation Tests

The CD spectrum analysis (Figure 6) showed that both purified proteins were folded. They adopted a mixed α/β secondary structure with a large proportion of beta structure in TTPBgp12 and with similar content of helical and beta structures in TFPgp17.

NanoDSF and backreflection technology was used to measure the stability, thermal unfolding and aggregation of TTPBgp12 and TFPgp17. Estimating the protein’s stability is very important to researchers studying these macromolecules. This quantification reveals information about their thermal and colloidal features. Consequently, it is possible to determine the optimal conditions for large-scale production and long-term storage of proteins. We have also performed stabilization experiments for proteins in the presence of different mono- and disaccharides that are potential binders. In the experiment, the most popular saccharide moieties, which build bacterial polysaccharides components, were used. Some of them are components of the outer core of the lipopolysaccharide (LPS) of *Y. enterocolitica* such as GalNAc, glucose, galactose [40]. Increasing the stability of the proteins in the presence of saccharide molecules could be the premise for complex formation with the protein. However, the nanoDSF experiment is a preliminary screen test and cannot support whether the proteins can interact with sugars. Our results present differences in the stability and aggregation of TTPBgp12 and TFPgp17 (Figure 7 and Figure 8, and Table 1).

TTPBgp12 was shown to have two unfolding events (at ~36 and 46 °C), suggesting that the protein is a dimer. We have observed that sugars overall were shown to have a different effect on the general stability of TTPBgp12, which aggregated in the presence of all the tested sugars. Interestingly, GlcNAc and GalNAc significantly increased the temperature at which the aggregation was detected, to ~56 and ~45 °C, by GlcNAc and GalNAc, respectively. It is worth noting that maltose and GlcNAc shift two unfolding events to one global unfolding. Interestingly, in the crystal structure of T7_TTPgp12 (the structural homolog of TTPBgp12), four domains were determined [13]. If we assume that TTPBgp12 adopts the spatial structure very similar to T7_TTPgp12, then it is possible that some domains unfold at ~36 and other domains unfold at ~46 °C. We think that the nozzle tip domain is the most distal and thus a separate domain. Therefore, it could unfold independently of the other domains.

TFPgp17 was shown to have a higher stability than TTPBgp12 with the one unfolding event at ~80 °C. Interestingly, the aggregation of TFPgp17 precedes the main unfolding event (at ~65 °C). It was observed that sugars can play various roles in the overall stability of TFPgp17. Among the sugars tested, maltose and GlcNAc were shown to have the highest effect on TFPgp17 aggregation. The sugars caused a large decrease in the observed temperature, to ~47 and ~56 °C, by maltose and GlcNAc, respectively. The presence of GalNAc increased the temperature of aggregation to ~80 °C, leading to the protein unfolding and aggregation events at the same time. NanoDSF and backreflection technology allowed us to determine the effect of the protein concentration on its stability (Figure 9 and Figure 10 and Table 2). The analysis of thermal unfolding profiles of the concentration gradient of TTPBgp12 and TFPgp17 clearly showed that with decreasing concentrations, both proteins increased in stability. As previously mentioned, the unfolding of individual domains of TTPBgp12 can even be detected at a concentration of ~0.25 mg/mL and higher. It is worth noting, that at the same time, a decrease in protein concentration can cause an increase in aggregation, for both TTPBgp12 and TFPgp17. These inverted patterns suggest that the optimization of the amount of protein in the buffer solution might help increase purification yield of the protein. It is more prone to aggregate when diluted, which is the same condition presented during the flow through the chromatography columns. Additionally, for long-term storage it might be more optimal to lower the aggregation under unfolded conditions and establish refolding conditions to keep higher concentrations of functional folded protein in the experimental assays.

### 3.4. Influence of TTPBgp12 and TFPgp17 on Biofilm Formation

As previously mentioned, tail phage proteins display a structural function, but they can play additional biological roles as well [20,27,28]. The purified TTPBgp12 and TFPgp17 were used in experiments regarding their effect on the bacterial *Yersinia enterocolitica* O:3 cells growth and biofilm formation. During bacteria incubation, TTPBgp12 or TFPgp17 was shown to have an inhibiting effect on bacterial culture growth. The effect was visible both in the bacterial suspension and during biofilm formation compared to the control samples (Figure 11).

Moreover, the visualization of biofilm layers stained with crystal violet confirmed this observation (Figure 12).

The obtained results showed that both TTPBgp12 and TFPgp17 were able to inhibit the growth of bacteria and could influence biofilm formation. This observation could open a wide field for further research towards explaining the mechanism of this phenomenon. In natural conditions, both TTPBgp12 and TFPgp17 were the part of the tail phage machinery involved in bacterial cells infection. The role of TTPBgp12 is still undetermined. However, it was shown that this protein can bind to bacterial cells after specific recognition by TFP. TTPBgp12 can limit bacterial communication with the external environment, decreasing the flow between the inside and the outside of the bacteria. In contrast, TFPgp17 was observed to be responsible for recognition and binding to the host receptors. It is most likely that the binding is reversible until the TTPAgp11 and TTPBgp12 come into action. Moreover, the bioinformatics analysis allows us to speculate that TFPgp17 itself could even be incorporated into the bacterial cell wall, resulting in the formation of pores. This could increase the cell membrane permeability of bacteria, efflux of a substances, and consequently even bacterial cell death. For now, these are our assumptions. Further research will allow us to find out the answers to our questions.

## 4. Conclusions

The goal of this work was to characterize the biochemical, functional structure, as well as the antibacterial activity of new members of the tail tubular protein B (TTPB) and tail fiber protein (TFP) families. We have chosen TTPBgp12 and TFPgp17, previously classified as tail phage proteins. The UniProt database was lacking on the information of the transcript, and homology levels of these proteins. Bioinformatic tools were used to analyze the similarity of the selected proteins to other members of their families. TTPBgp12 showed a high similarity of primary (69%) and spatial structure to the tail tubular protein of *Enterobacteria* phage T7.

TFPgp17 did not display a high similarity to other known TFPs. However, its N-terminal amino acid sequence is much more similar to corresponding fragments of other TFPs than its C-terminal regions. The C-terminus of TFPgp17 contains an intramolecular chaperone domain, which is common among the TFP family. The overall fold of the predicted 3D structure of TFPgp17 is very similar to S-layer proteins responsible for the stability of the bacteria cells.

For the first time, the genes of both proteins were overexpressed in *E. coli* cells and the proteins were purified by nickel-affinity and gel-filtration chromatography. The thermal unfolding experiments showed that TTPBgp12 is a two-domain protein in contrast to the single-domain protein, TFPgp17. TFPgp17 is a much more stable protein than TTPBgp12 which is indicated by the unfolding temperature of ~80 °C for TFPgp17, while the unfolding temperatures for the domains in TTPBgp12 are ~36 and 46 °C.

The presence of sugars affects the stability of both proteins causing their aggregation. It has also been observed that the stability of TTPBgp12 and TFPgp17 increases when of their concentrations are decreased. However, decreasing the protein concentration causes an increase in its aggregation, both for TTPBgp12 and TFPgp17. The incubation of TTPBgp12 or TFPgp17 with *Yersinia enterocolitica* O:3 inhibited bacterial cell growth, as well as impacted the biofilm formation. These results revealed the antibacterial activity of both tested phage proteins. The mechanism of this activity is not yet known. We hypothesize that TTPBgp12, which acts as an adhesive, may limit the uptake of nutrients in bacteria culture. TFPgp17, similar to other S-layer proteins, can form pores in the bacterial membrane, resulting in an increase in permeabilization and transport interruption. More studies are needed to further understand the structure and function of TTPBgp12 and TFPgp17. Our results suggest that both TTPBgp12 and TFPgp17 could potentially be used as a biofilm inhibiting factor.

## Figures and Tables

**Figure 1 molecules-25-04392-f001:**
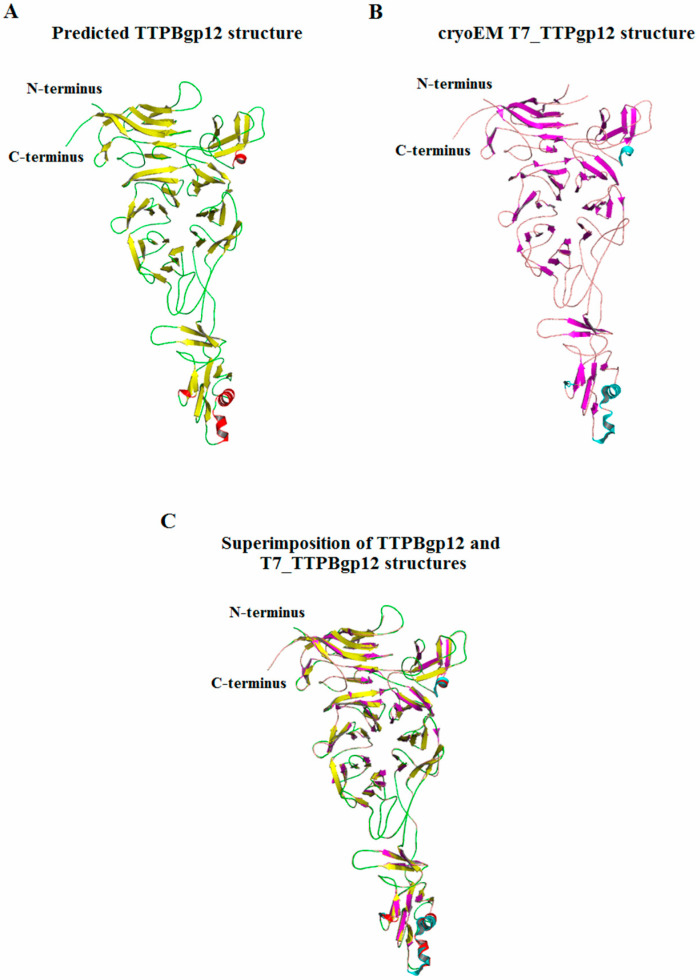
The predicted 3D structural model of tail tubular proteins B (TTPB)gp12 generated using the I-Tasser server [34]. (**A**) The presented structure was the best calculated model and was obtained using the cryoEM structure of T7_TTPgp12, (**B**) deposited in PDB (PDB code: 6r21). (**C**) The predicted structure of TTPBgp12 was superimposed onto cryoEM structure of the T7_TTPgp12 monomer.

**Figure 2 molecules-25-04392-f002:**
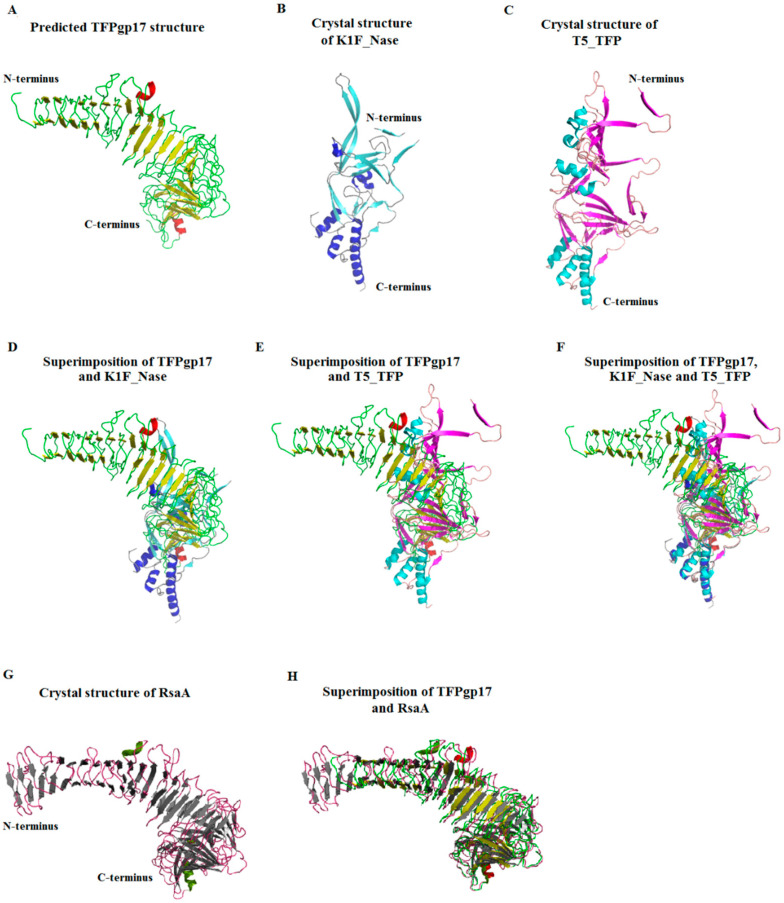
The predicted 3D structural model of TFPgp17 generated using the I-Tasser server [34]. (**A**) The presented structure was the best calculated model, and was obtained using the crystal structure of (**B**) endo-*N*-acetylneuraminidase of *Enterobacteria* phage φK1F, (K1F_Nase, PDB code: 3gw6) [18] and (**C**) L-shaped tail fiber protein of *Enterobacteria* phage T5 (T5_TFP, PDB code: 4uw8) [37]. The predicted structure of TTPBgp12 superimposed onto the crystal structure of K1F_Nase (**D**), T5_TFP (**E**) and both template proteins (**F**). (**G**) The crystal structure of RsaA (PDB code: 5n8p) [38], the structural analog of the predicted TFPgp17. (**H**) The predicted structure of TFPgp17 superimposed onto RsaA.

**Figure 3 molecules-25-04392-f003:**
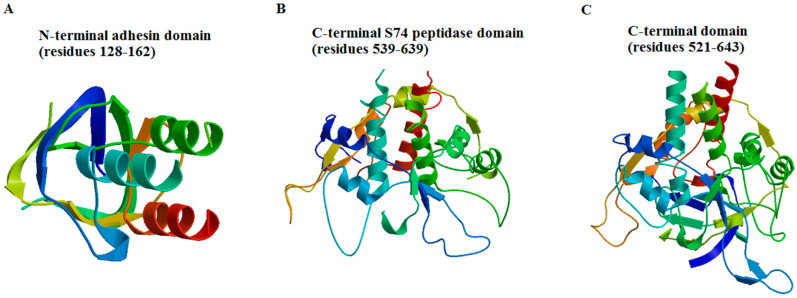
The predicted 3D partial structural models of tail fiber protein (TFP)gp17 generated using the Swiss-Model server [35]. (**A**) The model #1—N-terminal region consisting of the amino acid residues 128–162 build on the template of adhesin A (PDB code: 3d9x) [39]. (**B**) The model #2—the C-terminal S74 peptidase domain consisting of the amino acid residues 539–639 build on the template of the neck appendage protein (intramolecular chaperone) of *Bacillus* phage GA-1 (PDB code: 3gud) [18]. (**C**) The model #3—C-terminal region consisting of the amino acid residues 521–643 build on the template of the L-shaped tail fiber protein of *Enterobacteria* phage T5 (PDB code: 4uw8) [37].

**Figure 4 molecules-25-04392-f004:**
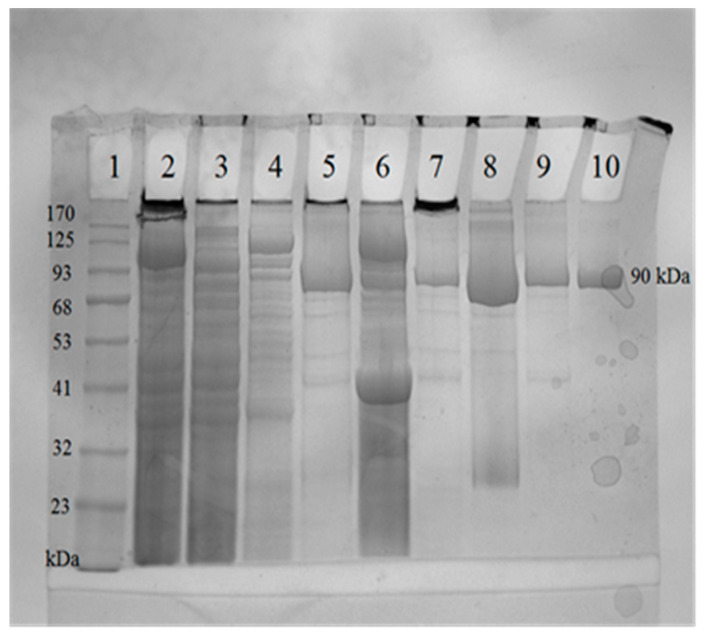
Analysis of TTPBgp12 from Yersinia phage φYeO3-12 using 12% SDS-PAGE. The lanes are as follows: (1) prestained protein ladder—mid-range molecular weight (10–180 kDa) (Nippon Genetics); (2) crude extract; (3) the flow through fraction and (4) pooled fractions eluted with 250 mM imidazole after the first round of nickel-immobilized affinity column; (5) TTPBgp12 fractions after Tobacco Etch Virus (TEV) protease cleavage; (6) pooled fractions eluted with 250 mM imidazole after the second round of Ni^2+^-affinity chromatography; (7) denaturated and undenaturated (8) TTPBgp12 fractions after ammonium sulfate precipitation; (9) denaturated and undenaturated (10) TTPBgp12 solution after gel-filtration chromatography.

**Figure 5 molecules-25-04392-f005:**
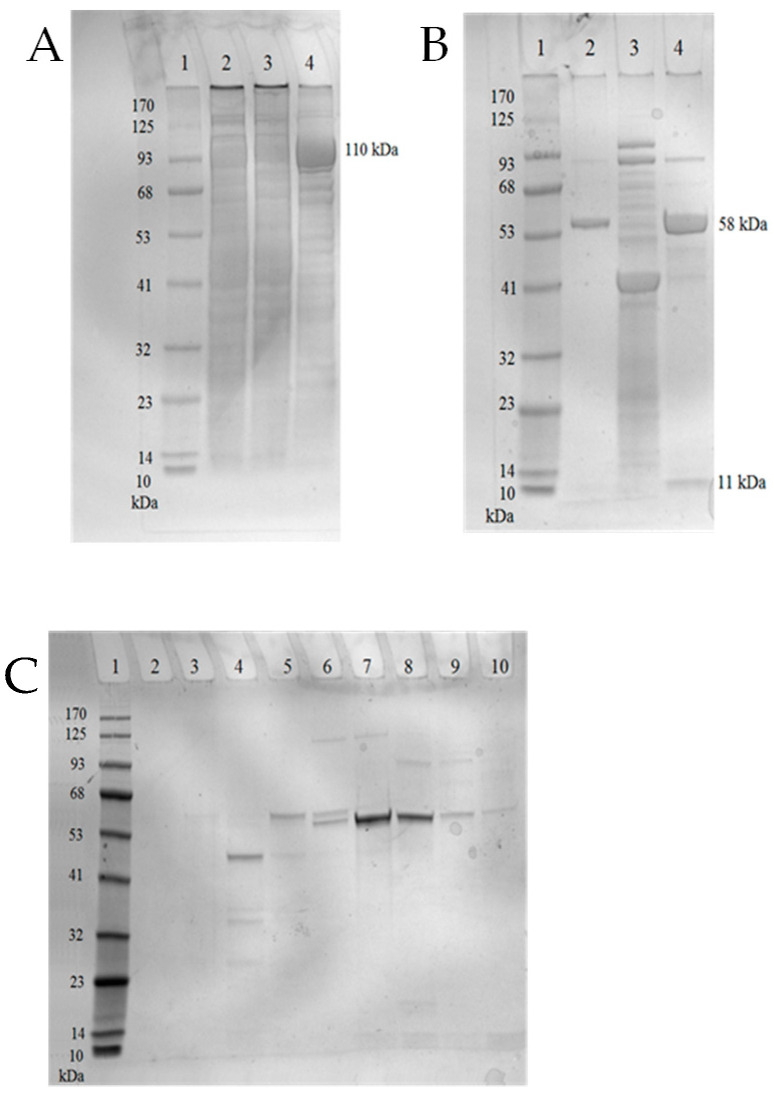
Analysis of TFPgp17 from *Yersinia* phage φYeO3-12 using 12% SDS-PAGE. (**A**) The first round of nickel-immobilized affinity column. The lanes are as follows: (1) prestained protein ladder—mid-range molecular weight (10–170 kDa) (Nippon Genetics); (2) crude extract; (3) the flow through fraction; (4) pooled fractions eluted with 250 mM imidazole. (**B**) The second round of nickel-immobilized affinity column. The lanes are as follows: (1) prestained protein ladder—mid-range molecular weight (10–170 kDa) (Abcam; (2) TFPgp17 fractions after TEV protease cleavage; (3) pooled fractions eluted with 250 mM imidazole; (4) TFPgp17 fractions after ammonium sulfate precipitation. (**C**) Fractions obtained after gel-filtration chromatography: (1) mid-range molecular weight (10–170 kDa) (ABCAM), (2) collected fraction 1, (3) collected fraction 2, (4) collected fraction 3, (5) collected fraction 4, (6) collected fraction 5, (7) collected fraction 6, (8) collected fraction 7, (9) collected fraction 8, (10) collected fraction 9.

**Figure 6 molecules-25-04392-f006:**
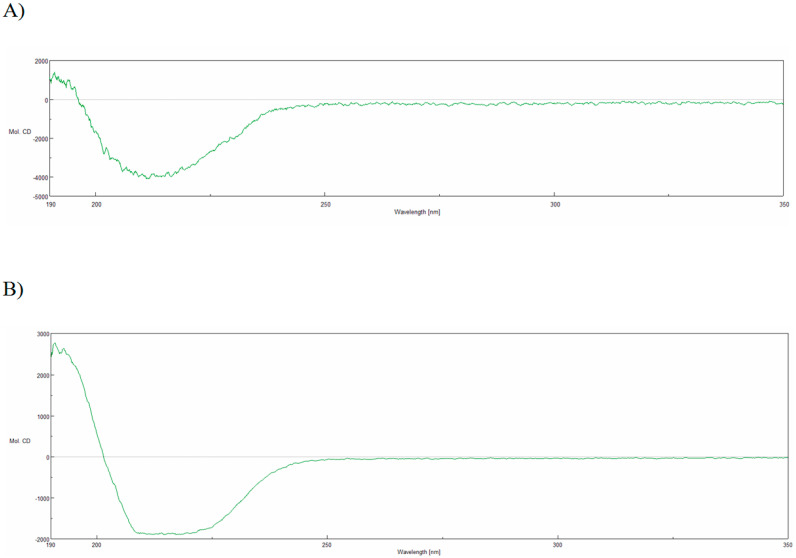
The circular dichroism (CD) spectrum of TTPBgp12 (**A**) and TFPgp17 (**B**) in units of molar ellipticity (deg cm^2^ dmol^−1^).

**Figure 7 molecules-25-04392-f007:**
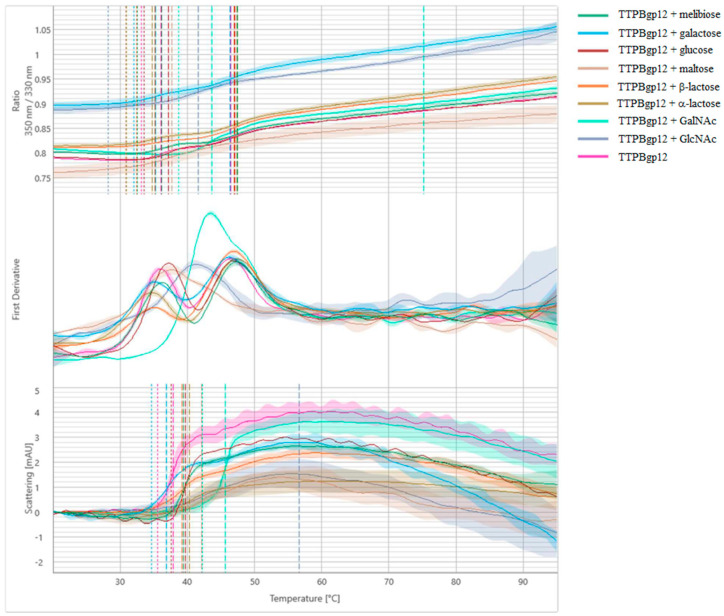
Melting analysis of TTPBgp12 with various sugars. The fluorescence ratio (350 nm/330 nm) is shown in the top panel, the first derivative is shown in the middle panel and scattering is shown in the bottom panel. Thermal unfolding and aggregation onsets (indicated as the vertical dotted lines), as well as unfolding and aggregation transitions (indicated as the vertical dashed lines), are indicated by vertical lines in the graph.

**Figure 8 molecules-25-04392-f008:**
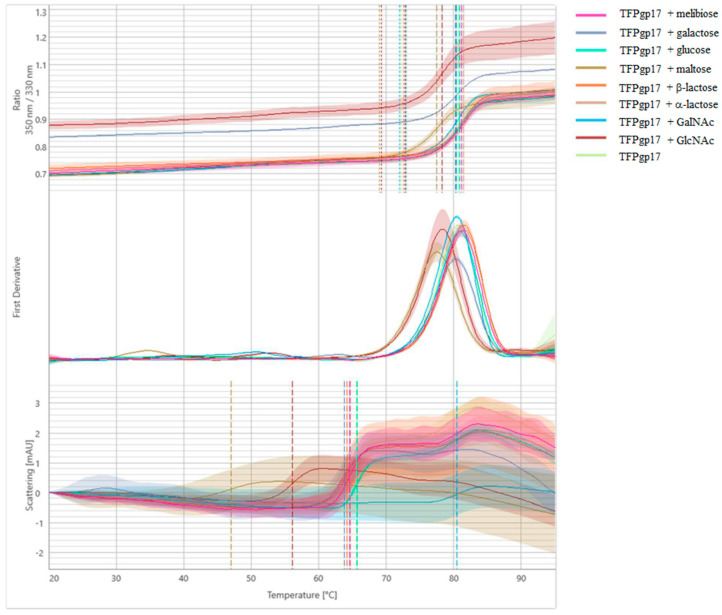
Melting analysis of TFPgp17 with various sugars. The fluorescence ratio (350 nm/330 nm) is shown in the top panel, the first derivative is shown in the middle panel and scattering is shown in the bottom panel. Thermal unfolding and aggregation onsets (indicated as the vertical dotted lines), as well as unfolding and aggregation transitions (indicated as the vertical dashed lines), are indicated by vertical lines in the graph.

**Figure 9 molecules-25-04392-f009:**
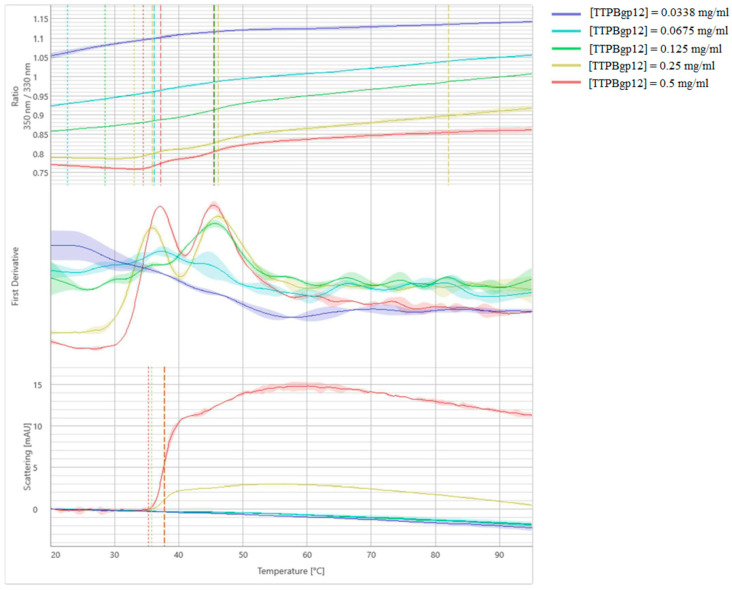
Melting analysis of the concentration gradient of TTPBgp12. The fluorescence ratio (350 nm/330 nm) is shown in the top panel, the first derivative is shown in the middle panel and scattering is shown in the bottom panel. Thermal unfolding and aggregation onsets (indicated as the vertical dotted lines), as well as unfolding and aggregation transitions (indicated as the vertical dashed lines), are indicated by vertical lines in the graph.

**Figure 10 molecules-25-04392-f010:**
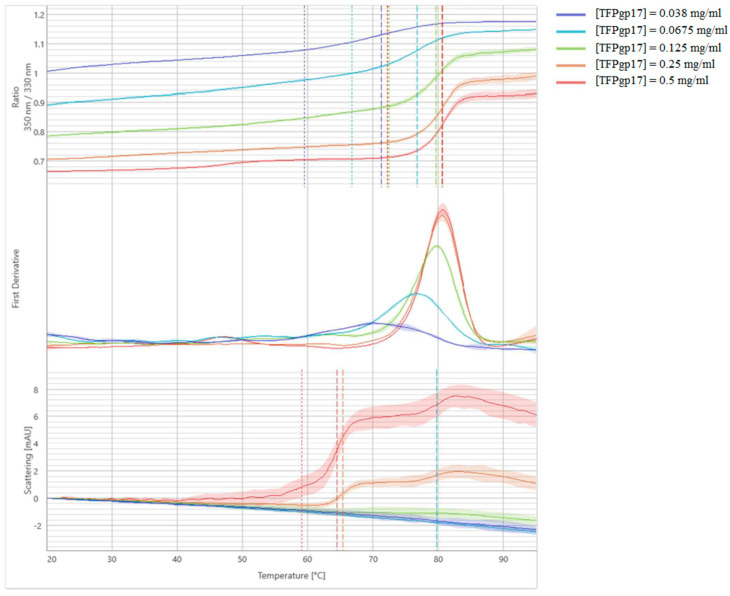
Melting analysis of the concentration gradient of TFPgp17. The fluorescence ratio (350 nm/330 nm) is shown in the top panel, the first derivative is shown in the middle panel and scattering is shown in the bottom panel. Thermal unfolding and aggregation onsets (indicated as the vertical dotted lines), as well as unfolding and aggregation transitions (indicated as the vertical dashed lines), are indicated by vertical lines in the graph.

**Figure 11 molecules-25-04392-f011:**
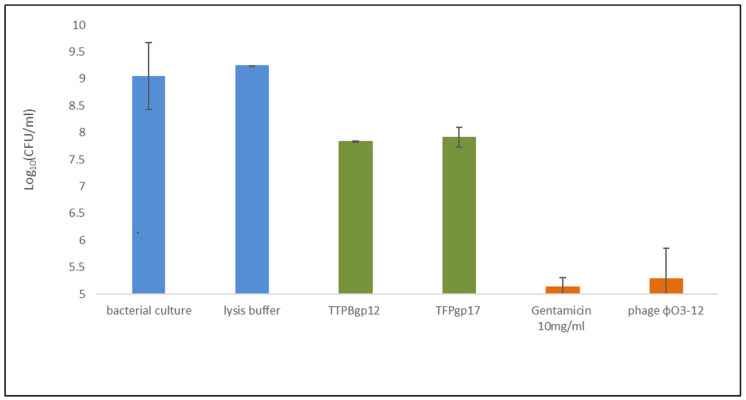
The effect of TTPBgp12 and TFPgp17 (in green) on *Yersinia enterocolitica* O:3 growth and the biofilm formation. The negative control (in blue) was lysis buffer (buffer in which proteins were suspended during the test) and the positive control (in orange) were gentamicin (10 mg/mL) and phage φYeO3-12. The untreated bacterial culture was shown in yellow. All results were presented as averages of results from three independent replicates in three parallel trials. Error bars represent the means, standard deviations. CFU, colony-forming unit.

**Figure 12 molecules-25-04392-f012:**
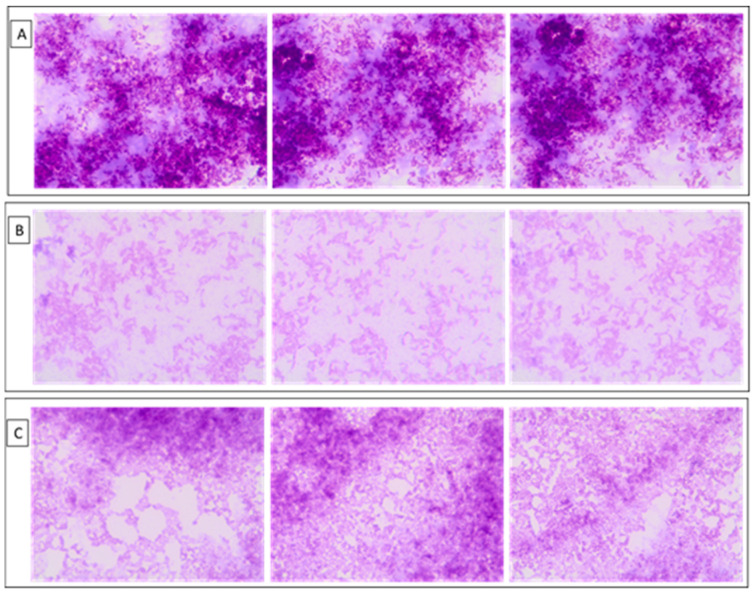
The effect of TTPBgp12 and TFPgp17 on the biofilm formation observed in a light microscope Olympus BX43 (magnitude 100×). The biofilm formed by *Yersinia enterocolitica* O:3 was stained with 1% crystal violet. (**A**) The control sample (*Yersinia enterocolitica* O:3 alone). (**B**) The bacteria incubated with TTPBgp12. (**C**) The bacteria incubated with TFPgp17.

**Table 1 molecules-25-04392-t001:** TTPBgp12 and TFPgp17 thermal unfolding and aggregation transitions midpoints (Tm (°C)) obtained from a melting scan of both proteins mixed with sugars, showing the fluorescence ratio 350 nm/330 nm), the first derivative onset and scattering.

Sample	Tm (°C)			
	Fluorescence Ratio 350 nm/330 nm		First Derivative	Scattering
	#1	#2		
TTPBgp12/melibiose	36.20	47.45	32.51	39.43
TTPBgp12/galactose	35.30	46.40	32.02	36.93
TTPBgp12/glucose	37.21	46.98	NA	33.76
TTPBgp12/maltose	37.64	NA	30.85	42.11
TTPBgp12/β-lactose	35.13	46.96	32.46	39.19
TTPBgp12/α-lactose	34.74	47.42	30.99	40.31
TTPBgp12/GalNAc	43.65	75.19	38.73	45.65
TTPBgp12/GlcNAc	41.66	NA	28.23	56.61
TTPBgp12	36.04	46.46	33.13	37.89
TFPgp17/melibiose	81.16	NA	72.83	64.64
TFPgp17/galactose	80.30	NA	72.88	63.78
TFPgp17/glucose	80.86	NA	72.89	65.71
TFPgp17/maltose	77.49	NA	69.03	46.99
TFPgp17/β-lactose	81.45	NA	72.57	64.19
TFPgp17/α-lactose	81.51	NA	73.01	64.66
TFPgp17/GalNAc	80.41	NA	71.96	80.45
TFPgp17/GlcNAc	78.29	NA	69.28	56.06
TFPgp17	80.70	NA	72.15	65.55

Each sample contained 0.25 mg/mL of the tail protein and 0.15 mg/mL of the sugar.

**Table 2 molecules-25-04392-t002:** TTPBgp12 and TFPgp17 thermal unfolding and aggregation transitions midpoints (Tm (°C)) obtained from a melting scan of concentration gradient of both proteins, showing the fluorescence ratio (350 nm/330 nm), the first derivative onset and scattering.

Sample		Tm (°C)			
		Fluorescence Ratio 350 nm/330 nm		First Derivative	Scattering
Protein	Concentration (mg/mL)	#1	#2		
TTPBgp12	0.0338	NA	NA	NA	NA
	0.0675	NA	NA	NA	NA
	0.125	45.58	NA	28.50	NA
	0.25	35.87	46.13	33.03	37.68
	0.5	37.17	45.46	34.46	37.79
TFPgp17	0.0338	71.30	NA	59.45	NA
	0.0675	76.80	NA	66.78	NA
	0.125	79.70	NA	72.49	NA
	0.25	80.54	NA	72.18	65.34
	0.5	80.67	NA	72.24	64.50

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
