# Peer review of "New Insights on the Feature and Function of Tail Tubular Protein B and Tail Fiber Protein of the Lytic Bacteriophage φYeO3-12 Specific for Yersinia enterocolitica Serotype O:3"

_molecules, 2020, doi:10.3390/molecules25194392_

Round 1
Reviewer 1 Report
The paper by Pyra et al is a well-documented study of two phage proteins that seem to have raised the interest of the research team because of the close relationship with other tail tubular proteins previously studied by themselves that were shown to perform functions additional to the mere structural ones.
The manuscript is methodologically correct, and the information supplied is complete, even excessive. Maybe the authors could consider reducing the body of the article and place some information under supplementary material (i.e. sequence alignments, E.F. analysis…).
I would also suggest using subheadings under Results and Discussion. The continuum in the text makes it hard to find the different subsections (alignments, modelling, purification, stability and unfolding…)
The English language is fine but needs revision. Some syntactic and punctuation minor errors can be detected. See, for instance, 129: Both, (no comma needed); 133 sentence needs rephrasing; 151, at the end: Than the slides (should be Then), etc. Please revise.
Author Response
We are very grateful for the comments and suggestions made by both reviewers. We have significantly improved the text. The manuscript was carefully checked by a native English speaker. I hope that the amendments that we have made will be satisfactory and that the revised manuscript will meet Macromolecules publication standards. Please find our response to criticisms and comments below. The inserted text is in red.
I am looking forward to hearing from you.
Yours sincerely,
Ewa Brzozowska
The paper by Pyra et al is a well-documented study of two phage proteins that seem to have raised the interest of the research team because of the close relationship with other tail tubular proteins previously studied by themselves that were shown to perform functions additional to the mere structural ones.
The manuscript is methodologically correct, and the information supplied is complete, even excessive. Maybe the authors could consider reducing the body of the article and place some information under supplementary material (i.e. sequence alignments, E.F. analysis…).
The authors have moved three Figures into Supplementary data. All changes were marked in red
I would also suggest using subheadings under Results and Discussion. The continuum in the text makes it hard to find the different subsections (alignments, modelling, purification, stability and unfolding…)
The authors subheading the subsections to Bioinformatics analysis, Gene cloning and protein overexpression, purification and analysis, Protein folding and aggregation tests
The English language is fine but needs revision. Some syntactic and punctuation minor errors can be detected. See, for instance, 129: Both, (no comma needed); 133 sentence needs rephrasing; 151, at the end: Than the slides (should be Then), etc. Please revise.
The authors, as well as a native speaker, revised the manuscript.
Reviewer 2 Report
The manuscript entitled “New insights on the feature and function of Tail Tubular Protein B and Tail Fiber Protein of the Lytic Bacteriophage φYeO3-12 specific for Yersinia enterocolitica Serotype O:3” by Anna Pyra and co-authors reported here the biochemistry and structural features of TTPBgp12 and TFPgp17 using bioinformatics predictions, how sugars affect the stability and aggregation of these proteins using in vitro assays, and demonstrated the antibacterial activity of these proteins using in vivo assays. Overall the experiments were designed properly and the conclusions are well supported by the experiment data. I have the following concerns which might improve the manuscript quality.
Major concern:
- The introduction of this manuscript was written in a way starting with bacteria then jump to the phage (which the authors care more in this manuscript). This should be reversed. Otherwise, the readers will think the manuscript is focused on the bacteria host, but not the phage.
- The authors tested sugars that can change the unfolding and aggregation properties of purified TTPBgp12 and TFPgp17. I guess the logic would be that the TFP protein could interact with LPS of bacteria host, that’s why these experiments have been performed. However, the authors did not emphasize this point, and of cause, this experiment also can not support that TFP can interact with sugars. This makes the observations of sugar can affect the aggregation of gp12 and gp17 be solitary data from the background the authors gave and other experiments the authors performed. And, also the molar concentrations of the sugars used in the experiments are also very high compared to proteins (100 fold higher?). I am wondering why the authors choose this concentration.
- The antibacterial activity of gp12 and gp17 is the major innovation of this manuscript. However, the authors did not give enough data to support this conclusion. The experiment design is also not suitable. The authors included a blank control, but not negative/positive controls. And, for the blank control, the authors used LB, that is not acceptable. LB is a nutrient-rich medium for bacteria growth, and this will potentially promote bacterial growth in the blank control. The best blank control would be the protein buffer which the gp12 and gp17 used in this experiment. The authors need to include negative and positive control. Negative control would be denatured gp12 or gp17, or any other inactive form of these proteins. Positive control would be an established protein (the best) or drug that has the antibacterial function.
Minor concerns:
- The resolution for all the figures is too low to read, this might due to the PDF conversion process, but this needs to be fixed.
- In the introduction, the authors say the gp12 is a two-domain protein. I guess this is based on the thermal unfolding experiments. The authors need to clearly say this two-domain feature is specifically associated with thermal unfolding. This is confusing since people would think gp12 has four domains.
- The manuscript has some typos that the authors need to fix. I included some, but there are more:
On page 4, line 144, “2.2 x 107 CFU/ml”, should be “2.2 x 107 (superscript)
CFU/ml”.
On pager 15, line 559, “decreasein the observed temperature,…”, may change to “decrease in the observed temperature,…”.
On page 16, line 568, “optimization of the amount of proteinin the buffer solution….”, may change to “optimization of the amount of protein in the buffer solution….”.
Author Response
We are very grateful for the comments and suggestions made by both reviewers. We have significantly improved the text. The manuscript was carefully checked by a native English speaker. I hope that the amendments that we have made will be satisfactory and that the revised manuscript will meet Macromolecules publication standards. Please find our response to criticisms and comments below. The inserted text is in red.
I am looking forward to hearing from you.
Yours sincerely,
Ewa Brzozowska
The manuscript entitled “New insights on the feature and function of Tail Tubular Protein B and Tail Fiber Protein of the Lytic Bacteriophage φYeO3-12 specific for Yersinia enterocolitica Serotype O:3” by Anna Pyra and co-authors reported here the biochemistry and structural features of TTPBgp12 and TFPgp17 using bioinformatics predictions, how sugars affect the stability and aggregation of these proteins using in vitro assays, and demonstrated the antibacterial activity of these proteins using in vivo assays. Overall the experiments were designed properly and the conclusions are well supported by the experiment data. I have the following concerns which might improve the manuscript quality.
Major concern:
- The introduction of this manuscript was written in a way starting with bacteria then jump to the phage (which the authors care more in this manuscript). This should be reversed. Otherwise, the readers will think the manuscript is focused on the bacteria host, but not the phage.
The authors reversed the parts describing the bacteria and the phage according to the Reviewer’s suggestion. The changes are in red color.
- The authors tested sugars that can change the unfolding and aggregation properties of purified TTPBgp12 and TFPgp17. I guess the logic would be that the TFP protein could interact with LPS of bacteria host, that’s why these experiments have been performed. However, the authors did not emphasize this point, and of cause, this experiment also can not support that TFP can interact with sugars. This makes the observations of sugar can affect the aggregation of gp12 and gp17 be solitary data from the background the authors gave and other experiments the authors performed. And, also the molar concentrations of the sugars used in the experiments are also very high compared to proteins (100 fold higher?). I am wondering why the authors choose this concentration.
In this paper, the authors focused on the characterization of new phage tail proteins concerning their primary, secondary, and tertiary structure (predicted structure) using bioinformatics tools. The authors also assumed (based on the previous study) that both proteins can bind and/or hydrolysate sugar substrates. NanoDSF was used as a screening method to check whether some saccharides increase the stability of the proteins that could mean they form the complex.
The Authors emphasized this point in the selection label, Discussion as follows: In the experiment, the most popular saccharide moieties building bacterial polysaccharides components were used. Some of them are components of the outer core of the lipopolysaccharide (LPS) of Y. enterocolitica such as GalNAc, glucose, galactose. Increasing the stability of the proteins in the presence of saccharide molecules could be a premise for complex formation with the protein. However, the nanoDSF experiment is a preliminary screen test and cannot support that the proteins can interact with sugars.
Increasing sugar to protein ratio leads to increased protein stability up to a saturation limit supports the water replacement theory. In the literature, the molar ratio of sugar to a protein of around 350 - 400 was shown to be sufficient to stabilize freeze- and spray dried monoclonal antibodies. In case of liquid solution, we decided to use a smaller molar ratio (150 mM) than dry-freeze. The authors used the same conditions as were described in the previous paper by Pyra et al (2020). The saccharide concentration was established in the preliminary studies performed by NanoTemper Company.
- The antibacterial activity of gp12 and gp17 is the major innovation of this manuscript. However, the authors did not give enough data to support this conclusion. The experiment design is also not suitable. The authors included a blank control, but not negative/positive controls. And, for the blank control, the authors used LB, that is not acceptable. LB is a nutrient-rich medium for bacteria growth, and this will potentially promote bacterial growth in the blank control. The best blank control would be the protein buffer which the gp12 and gp17 used in this experiment. The authors need to include negative and positive control. Negative control would be denatured gp12 or gp17, or any other inactive form of these proteins. Positive control would be an established protein (the best) or drug that has the antibacterial function.
The authors repeated the experiment. They used gentamicin as a positive control, although the mechanism of action of the antibiotic differs from the mechanism of the phage proteins. Gentamicin kills the bacteria while phage proteins slow down bacterial multiplication. The concentration of the antibiotic in the experiment was 2 µg/ml (MIC90). As a negative control, the authors used adhesin of the T4 phage. This protein doesn't bind to the polysaccharide structures of the Yersinia strain. Since both TPBgp12 and TFPgp17 proteins aggregate after denaturation, they could not be used as the negative control.
The additional results ( the negative and the positive control) have been added into the main text
Minor concerns:
- The resolution for all the figures is too low to read, this might due to the PDF conversion process, but this needs to be fixed.
The figures have been improved
- In the introduction, the authors say the gp12 is a two-domain protein. I guess this is based on the thermal unfolding experiments. The authors need to clearly say this two-domain feature is specifically associated with thermal unfolding. This is confusing since people would think gp12 has four domains.
The authors clarified this issue.
- The manuscript has some typos that the authors need to fix. I included some, but there are more:
On page 4, line 144, “2.2 x 107 CFU/ml”, should be “2.2 x 107 (superscript)
CFU/ml”.
It has been corrected
On pager 15, line 559, “decreasein the observed temperature,…”, may change to “decrease in the observed temperature,…”.
I It has been corrected
On page 16, line 568, “optimization of the amount of proteinin the buffer solution….”, may change to “optimization of the amount of protein in the buffer solution….”.
It has been corrected
Round 2
Reviewer 2 Report
The authors addressed most of the issues that I was concerned about, except for one major concern and one minor concern. The minor concern is still the second one in my previous comments about the two-domain issue of gp12. The authors said this has been clarified, but I do not see the change in the introduction. The major concern is the blank control that the authors used to demonstrate the antibacterial functions of gp12 and gp17. As I mentioned previously, the authors can not use LB. If PBS is the buffer used for the protein samples, just add 10 uL PBS as a blank control. The authors do not want to add anything that is not present in the protein samples, negative control, and positive control. And, the authors have to add the same volume of blank, positive, negative, gp12, and gp17 to the bacteria culture. In this case, for the blank control, just add 10 uL PBS, if the authors used PBS buffer to dissolve the gp12, gp17, negative and positive controls. If not, the authors have to change all controls to the same buffer that the blank control used, and re-do all the controls and samples. The idea is the authors have to exclude the possibility that the antibacterial effects of gp12 and gp17 are not caused by anything else in the protein samples, thus using proper controls is essential. Ideally, the authors need to show the results in both the bar graph (Figure 11) and the staining (Figure 12).
Author Response
Dear Reviewer,
Thank you for your comments and suggestions. We supplemented the text (in red) with additional results, there was a misunderstanding between the authors, and the errors resulted from an editorial mistake. We hope you will be satisfied with the minor revision corrections.
Reviewer comments:
- The antibacterial activity of gp12 and gp17 is the major innovation of this manuscript. However, the authors did not give enough data to support this conclusion. The experiment design is also not suitable. The authors included a blank control, but not negative/positive controls. And, for the blank control, the authors used LB, that is not acceptable. LB is a nutrient-rich medium for bacteria growth, and this will potentially promote bacterial growth in the blank control. The best blank control would be the protein buffer which the gp12 and gp17 used in this experiment. The authors need to include negative and positive control. Negative control would be denatured gp12 or gp17, or any other inactive form of these proteins. Positive control would be an established protein (the best) or drug that has the antibacterial function.
The authors performed an additional experiment. They used gentamicin and bacteriophage φYe O3-12 specific to the tested bacterial strain, as a positive control, although the mechanism of action of the antibiotic differs from the mechanism of the phage proteins. Gentamicin kills the bacteria while phage proteins slow down bacterial multiplication. The concentration of the antibiotic in the experiment was 1 mg/ml. As a negative control, the authors used a buffer in which the tested protein was suspended. Since both TPBgp12 and TFPgp17 proteins aggregate after denaturation, they could not be used as the negative control. Control with untreated bacteria in LB medium is also important to show the differences in the effectiveness of individual proteins, despite this, I agree with the reviewer that a buffer control should be added, therefore these results were included in the graph.
The additional results ( the negative and the positive control) have been added into the main text
- In the introduction, the authors say the gp12 is a two-domain protein. I guess this is based on the thermal unfolding experiments. The authors need to clearly say this two-domain feature is specifically associated with thermal unfolding. This is confusing since people would think gp12 has four domains.
In the abstract, the authors say the gp12 is a two-domain protein. To clarify this, they added: “Based on the thermal unfolding experiment, TTPBgp12 seems to be a two-domain protein that aggregates in the presence of sugars such as maltose and N-acetylglucosamine (GlcNAc)”.
Best Regards
Ewa Brzozowska, PhD, DSc
Laboratory of Medical Microbiology
Institute of Immunology and Experimental Therapy,
Polish Academy of Sciences
Weigl 12
53-114
Wroclaw